# Neural Correlates of Numerical Estimation: The Role of Strategy Use

**DOI:** 10.3390/brainsci12030357

**Published:** 2022-03-07

**Authors:** Sarit Ashkenazi, Refael Tikochinski, Dana Ganor-Stern

**Affiliations:** 1Learning Disabilities, The Seymour Fox School of Education, The Hebrew University of Jerusalem, Jerusalem 9190501, Israel; 2Technion, Israel Institute of Technology, Haifa 3200003, Israel; refael.tikochinski@mail.huji.ac.il; 3Achva Academic College, Shikmim 7980400, Israel; danaganor@gmail.com

**Keywords:** computation estimation, math strategy, functional magnetic resonance imaging, approximated calculation, parietal lobule

## Abstract

Introduction: Computation estimation is the ability to provide an approximate answer to a complex arithmetic problem without calculating it exactly. Despite its importance in daily life, the neuronal network underlying computation estimation is largely unknown. Methods: We looked at the neuronal correlates of two computational estimation strategies: approximated calculation and sense of magnitude (SOM)–intuitive representation of magnitude, without calculation. During an fMRI scan, thirty-one college students judged whether the result of a two-digit multiplication problem was larger or smaller than a given reference number. In two different blocks, they were asked to use a specific strategy (AC or SOM). Results: The two strategies activated brain regions related to calculation, numerical cognition, decision-making, and working memory. AC more than SOM elicited activations in multiple, domain-specific brain regions in the parietal lobule, including the left SMG (BA 40), the bilateral superior parietal lobule (BA 7), and the right inferior parietal lobule (BA 7). The activation level of the IFG was positively correlated to individual accuracy, indicating that the IFG has an essential role in both strategies. Conclusions: These finding suggest that the analogic code of magnitude is more involved in the AC than the SOM strategy.

## 1. Introduction

### 1.1. Brain and Calculation

In the last decades, brain-imaging techniques, such as functional magnetic resonance (fMRI), have made it possible to understand the neural mechanisms underlying basic cognitive functions (such as language) and, specifically, basic numerical processes (such as numerical comparison or simple arithmetic) [1,2,3]. It is well accepted that the representation of approximate quantities (ANS; approximate number system) is a foundational preverbal ability that is supported by the intraparietal sulcus (IPS) in the posterior parietal cortex [4,5,6,7]. Despite its importance in daily life, much less is known about the neuronal network underlying computation estimation, which is the ability to provide an approximate answer to a complex arithmetic problem without calculating it exactly [8]. This ability is necessary when exact calculation is impossible due to time or attentional limitations.

### 1.2. Brain–Exact vs. Approximate

A series of neuroimaging studies have focused on the question of whether there is dissociation between exact and estimation processes. While exact processes are verbally based and learned, arithmetic estimation is more heavily dependent upon preverbal innate representation of magnitude, supported by the IPS. In line with this view, Dehaene et al. [9] were the first to provide behavioral and neural evidence for such a dissociation using simple, single-digit numbers. The greatest difference in favor of approximation was found in the bilateral inferior parietal lobule. Conversely, the greatest difference in favor of exact calculation was found in the left inferior prefrontal cortex, with a smaller focus in the left angular gyrus, two areas associated with language processes. In line with the strong IPS involvement in estimation, a recent neuropsychological study tested the performance of a patient with damage to the IPS on the computation estimation task. The abnormal patterns shown by this patient (i.e., absence of distance and size effects) point to the importance of the IPS in approximation [10].

However, only a few studies have cited the above dissociation between approximation and exact calculation (i.e., involvement of preverbal representation of quantity in approximation and involvement of verbally mediated quantity representation in exact estimation) [11,12,13], while others have not reported similar results [14]. Specifically, the neuroanatomical dissociations between exact and approximate calculations have not been replicated in different study designs or by other [13,15,16] studies. For example, [15] tested exact and approximate single-digit addition problems in children and adults and found greater activations in approximation in frontal and parietal brain areas, with lack of significant activations in the contrast of exact more than approximation [16]. Similarly, an fMRI study tested bilingual adults in exact and approximate complex addition problems and discovered significantly higher levels of activation in approximate addition compared with exact addition in the left inferior frontal, middle frontal, and inferior parietal regions, with no results for the contrast of exact more than approximation [16]. The foregoing demonstrates that multiple differences are observed between exact and approximation rather than involvement or lack of involvement of verbally mediated arithmetic.

Finally, ref. [13] looked at complex approximate and exact arithmetic (including addition, subtraction, multiplication, and division of integers or fractions). Approximate arithmetic relative to exact computation elicited greater activation in the bilateral IFG, middle temporal gyrus, angular gyrus, and dorsomedial prefrontal cortex. In contrast, exact computation elicited greater activations in the left Rolandic operculum and bilateral hippocampus.

### 1.3. Computation Estimation

Computation estimation is a form of approximation and should therefore activate the preverbal coding of quantity (semantic code) based upon the IPS. Different tasks have been used to investigate computation estimation. In one paradigm, participants provided an estimated answer for a given arithmetic problem [17,18,19]. In this task, participants used mainly rounding procedures [17,18,19]. Participants across age groups were shown to choose which strategy for which item adaptively (two-digit numbers with unit digits smaller than 5 rounded down and those with units digits larger than 5 rounded up to the nearest ten) [18,20,21]. In another experimental paradigm, participants were presented with simple arithmetic problems accompanied with two answers, and they were asked to select the more plausible choice [9,12,13,14].

In the estimation comparison task used in the current study, participants are presented with a two-digit multiplication problem together with a reference number, and they are asked to estimate whether the exact answer is larger or smaller than the reference number. The exact answer could be far or close from the reference number, and it could be larger or smaller than it. Across studies, performance was enhanced, in terms of speed and accuracy, for trials in which the exact answer was larger (vs. smaller) than the reference number (indicating a size effect) and when it was far (vs. close) from the exact answer (indicating a distance effect) [22,23,24,25]. Similar patterns of size and distance effects in computation estimation tasks were also found in other studies [26].

Two strategies were used to solve this task [22,23,24,25]: (1) The approximated calculation (AC) strategy involves rounding either one or two multiplicands, multiplying the rounded numbers, and comparing the product to the reference number. The use of such rounding procedures has been shown in past research when participants produced estimates for multi-digit addition [18] or multiplication [23,24,25,27] problems. Note that in this task, this strategy was not separated into rounding-down and rounding-up strategies. (2) The sense of magnitude (SOM) strategy is unique to the current estimation task, and it is based on unrefined, intuitive approximated representations of magnitude for the results of such problems, probably based on the ANS. Across studies, it has been found that the SOM strategy produces faster but less accurate responses, while the calculation strategy produces slower but more accurate responses [22,23,24,25].

Adults as well as children used both strategies and selected which strategy to use in an adaptive manner. Specifically, the SOM strategy was used more often in the far condition (the reference number was far from the correct answer), and the time-consuming approximated calculation strategy was used more often in the close condition (the reference number was close to the correct answer), where the SOM could not guarantee a correct response [22,23,24,25].

Most past research that investigated strategy use and its link to accuracy and speed allowed participants to choose the strategy freely to use for each item. This created confounding of strategy use and strategy choice. The choice-no-choice paradigm was used to provide independent information about strategy use independent of strategy choice [28,29]. This method is composed of two no-choice blocks and a single choice block, which are composed of parallel items. In the no-choice condition, before the beginning of each block, participants are instructed which strategy to use, and they are to use this strategy solely throughout this block. Thus, performance in these no-choice blocks provides measures of strategy execution only without strategy choice, as the same strategy was used throughout the block. In the choice condition, participants are allowed to choose which strategy to use in each item (as in previous studies with this paradigm). The results of this block provide information about strategy choice and the extent to which it is linked to strategy execution. The speed and accuracy data in this block are expected to replicate the patterns found in past studies in which participants were allowed to choose which strategy to use.

Only one study so far has tested the neural mechanisms underlying strategy selection and strategy execution in computational estimation of multiplication problems [8]. The strategies were rounding down and rounding up the multiplicands. Participants could either select which strategy to use (choice condition), or they were instructed which strategy to use (no-choice condition). The main finding of this study was related to the difference in brain activation between the choice and no-choice conditions. Neuroimaging data showed greater brain engagement in the inferior parietal cortex (i.e., right and left supramarginal gyrus (SMG), right angular gyrus (AG), and right precuneus), frontal cortex (right dorsolateral prefrontal cortex (DLPFC) and right and left middle frontal gyrus (MFG)), and anterior cingulate cortex (ACC) in the choice rather than in the no-choice conditions. There were no greater brain activities in the no-choice condition relative to the choice condition. Additionally, rounding-up strategy was associated with more parietal cortex change in neural activity than the rounding-down strategy.

### 1.4. Present Study

The contribution of the present study is two-fold. First, to the best of our knowledge, this is the only study using brain imaging that examined the neural mechanisms underlying the two strategies used to estimate the results of multi-digit multiplication problems: the AC strategy that involves rounding and calculation procedures and the SOM strategy that relies on intuition about the approximate magnitude of the results. The only study that examined the neural mechanism for such problems focused on the difference between choice and no-choice conditions [8]. Second, it provides information about the neuronal basis for use of the different strategies in this task. Importantly, the information it provides about strategy use is not confounded by strategy choice due to the use of the choice-no choice paradigm [28]. To the best of our knowledge, the only study that examined the neurological basis of strategy use when estimating the results of such problems was conducted on a neurological patient with focal damage in the IPS. The results of this patient were unique, as she did not show the standard distance and size effects that are fundamental to this task and numerous others. This result indicates the importance of the IPS in this estimation task [10].

In the present study, during an fMRI scan, participants judged whether the results of two-digit multiplication problems are larger or smaller than given reference numbers [22]. Using the no-choice paradigm [29], in one block, participants were instructed to use the AC strategy and in the other to use the SOM strategy. The blocks were composed of parallel stimuli as was done in [30].

We believe that (1) since computation estimation requires activation of the ANS and requires calculations, computation estimation across strategies will engage the fronto-parietal network [29]. (2) The SOM more than AC involves intuitive understanding of quantities and ANS representations, and both should engage the IPS [4,5,6,7]. Hence, in the SOM block more than in the AC block, greater engagement will be found in the bilateral parietal lobule, especially the IPS. Conversely, (3) the AC block compared to SOM involves more left lateralized verbal representation of numbers and more mental effort requiring executive functions. Therefore, the greatest difference in favor of the AC block compared to the SOM block will be found in the left inferior prefrontal cortex.

## 2. Materials and Methods

Twenty-four participants took part in the experiment, and all of them were college students. The mean age of the participants was 23.6 (S.D. 2.09) years old, and 10 of them were females. All participants received monetary compensation for taking part in the study. Participants gave written informed consent to participate in the study, which was previously approved by the Institutional Review Board (IRB). This study was carried out following the guidelines of the protocol approved by the IRB. Five participants were excluded from the analysis due to movement greater than 2 mm during the scan.

### 2.1. Brain Imaging

#### 2.1.1. Procedure and Stimuli

##### Basic Task

The experiment was conducted individually. The task employed in this study is the estimation comparison task used by Ganor-Stern [22]. In each trial, a 2D multiplication problem appeared on the screen with a reference number below it. Participants were asked to estimate whether the answer for each problem was larger or smaller than the reference number. They had to press the “right” key if they estimated it to be larger than the reference number and the “left” key if they estimated it to be smaller. Participants were explicitly told that they should not solve the problems exactly but should only estimate whether the answer was larger or smaller than the given number. The numbers remained on the screen until participants responded. The order of trials was random.

The experiment was composed of 4 blocks. Block 1 was conducted outside the scanner. The purpose of this block was to provide training on the task that was later conducted within the scanner and to obtain trial by trial information on the strategies used through self-reports. At the beginning of Block 1, the experimenter explained the experimental task, and participants were given two examples to make sure that they understood the task. Then, they were given 8 trials for which they responded by keypress alone. In the next 16 trials, they were asked after they responded to each trial to describe how they reached their answer. At the end of the block, participants were told by the experimenter that, from past research, we know that people solve this task mainly by two strategies: either by using rounding procedures or by using a more intuitive sense of magnitude. The experimenter explained that the next blocks would be conducted inside the scanner, and at the beginning of the block, the experimenter would indicate to each participant which strategy to use.

Blocks 2–4 were conducted inside the scanner. In these blocks, participants responded by keypress only. Each block was composed of 40 trials. At the beginning of each block, participants were instructed through the microphone which strategy to use. In Block 2, they were told that they could use any strategy they want (the choice block). In Blocks 3 and 4 (the no-choice blocks), participants were instructed to use either the AC strategy or the SOM strategy throughout the block. The order of Blocks 3 and 4 were counterbalanced across participants. The results of only Blocks 3 and 4 were analyzed and are reported in the current study.

At the beginning of each block, a blank slide was shown for 30 s and at the end of each block for 60 s. Each trial contained a 2-digit × 2-digit multiplication problem with a target number below the operation, and participants were asked to indicate by a keypress whether the exact solution was larger or smaller than the target number. The problem and the target number appeared for 12 s. After each trial, there was a jitter alternation between 10, 12, or 16 s randomly. Each trial in our fMRI protocol was set for 12 s from the appearance of the multiplication problem until that problem disappeared.

At the beginning of the experiment, participants signed an informed consent statement.

#### 2.1.2. fMRI Data Acquisition

Whole-brain functional data were acquired using a 3-Tesla Philips ingenia MRI scanner using a gradient echo planar imaging (EPI) sequence (TR = 2 s; TE = 35 ms; flip angle = 90 degrees). The order of imaging acquisition was ascending–interleaved, covering the entire brain of participants. For each functional volume, 33 slices (3 mm thickness, FOV = 230 mm × 245 mm × 115 mm matrix = 76 × 83) were collected, resulting in a spatial resolution of 3 mm isotropic voxels. Each scanning session included the acquisition of a T1-weighted three-dimensional volume (voxel dimension = 1 mm × 1 mm × 1 mm) for co-registration and anatomical location of functional data.

#### 2.1.3. fMRI Preprocessing

Functional MRI data were analyzed using the BrainVoyager V21.4.5 software (https://www.brainvoyager.com/index.html, accessed on 1 March 2022). In the preprocessing stage, we first implemented slice scan-time correction with respect to the first slice, using cubic spline interpolation. Secondly, we applied a high-pass temporal filter on the data using GLM approach with Fourier basis set of 2 sine/cosine pairs. Then, 3D motions were then detected using trilinear interpolation and were corrected via sinc interpolation. Five subjects were excluded in this stage due to extreme head movements inside the magnet (>2 mm). Finally, after aligning the functional images to the anatomical image, we normalized the brain images to Talairach coordinates space and spatially smoothed the data with a 6 mm full-width half-maximum Gaussian kernel to decrease spatial noise prior to statistical analysis.

#### 2.1.4. Statistical Analysis

General linear model (GLM) was conducted first at the individual participants level, yielding a unique activation map for each participant. Those activation maps then were analyzed together in the second-group-level analysis in order to create a voxel-wise t-statistics map. We built three different statistical maps corresponding to the following contrasts: AC block vs. rest; SOM block vs. rest; and AC block vs. SOM block. For dealing with multiple comparisons, we defined a stringent statistical signification threshold (*p* < 0.005) with FDR cluster-based correction; (please see the same contrast with a higher threshold (*p* < 0.001) in the Appendix A).

## 3. Results

### 3.1. Behavioral Analysis

Reaction times. Reaction times (RT) of all trials in each block were collected and averaged across participants. The mean RT was shorter for the SOM block than for the AC block (for the SOM block: M = 1.92, SD = 0.48; for the AC block: M = 3.27, SD = 0.75, t(18) = 7.98, *p* < 0.001).

Accuracy. The mean accuracy rate was higher for the AC block than for the SOM block (SOM block: M = 0.91, SD = 0.05; AC block: M = 0.95, SD = 0.07; t(18) = 2.20, *p* < 0.05). The order of the blocks did not affect the RTs or accuracy (*p* > 0.05 for any block).

### 3.2. fMRI Analysis

BOLD activations for corrected answers were collected and analyzed via first- and second-level linear-models (for more information, see the Method section). In the first stage, for each block, we explored the brain areas that were significantly activated (*p* < 0.005 with cluster-based correction) compared to rest.

#### 3.2.1. AC Strategy

AC was associated with changes in neural activity in the frontal cortex, specifically the middle and inferior frontal gyrus (bilateral MFG and left IFG; BA 9, 46 and 6), the left superior frontal gyrus (BA 8), and the insula; in the posterior parietal cortex, including the intraparietal sulcus (bilateral IPS; BA 7, 19, and 40); in the ventral-visual stream, including the fusiform gyrus (BA 37 and 18); in the bilateral cuneus (BA 7); and in the left extra-nuclear (BA 30). Significant deactivations were found in the right middle temporal gyrus (BA 22) and in the left medial frontal gyrus (BA 10) (see Table 1 and Figure 1A and Appendix A for *p* < 0.001).

#### 3.2.2. SOM Block

SOM was associated with changes in neural activity in the frontal cortex, including the middle and superior frontal gyrus (BA 10 and 6) and the right insula (BA 13); in the posterior parietal cortex, including the right SMG (BA 40) and the left inferior parietal lobule (IPL) (BA 40 and 7); in the right and left posterior cingulate cortex (BA 23); in the occipital lobule, specifically the inferior and the lateral occipital gyrus (BA 18 and 37); and in the left inferior temporal gyrus (BA 21) (see Table 2 and Figure 1B and Appendix A for *p* < 0.001).

#### 3.2.3. Differences between the Strategies

In this stage, we compared the differences directly in the brain activation between the strategies. A larger, cluster-based activation (*p* < 0.005 with correction, please see *p* < 0.001 results in the Appendix A) was found for the AC-strategy (relative to the SOM-strategy) in the frontal cortex, including the bilateral IFG (BA 6 and 9), right superior and middle frontal gyrus (BA 6), left orbital gyrus (BA 10), right insula, and the precentral gyrus (BA 6); in the parietal lobule, including the left SMG (BA 40), the bilateral superior parietal lobule (BA 7), and the right IPL (BA 7); in the occipital lobule, including the right middle occipital gyrus (BA 17), left inferior occipital gyrus (BA 18), left lateral occipitotemporal gyrus (BA 37), and the right cuneus (BA 7); in the bilateral thalamus; and bilateral cingulate gyrus (BA 32 and 24) (see Table 3 and Figure 2). For the SOM-strategy, we found a larger engagement (relative to the AC strategy; *p* < 0.005 with cluster-based correction, please see *p* < 0.001 results in the Appendix A) in the frontal cortex, specifically in the bilateral superior and inferior frontal gyrus (BA 6, 8, 45, and 46); in the parietal lobule, including the right postcentral gyrus (BA 4), the left precuneus (BA 31), and the right SMG (BA 39); in the occipital lobule, including the bilateral superior occipital gyrus (BA 18), right inferior occipital gyrus (BA 37), bilateral lingual gyrus, and the left cuneus (BA 30); in the temporal lobule, including bilateral middle and superior temporal gyrus (BA 21 and 22) and the left inferior temporal gyrus (BA 21); and the right cingulate gyrus (BA 31) (see Table 3 and Figure 2 and Appendix A for *p* < 0.001).

#### 3.2.4. Region of Interest Analysis

Two out of the six pre-defined ROIs (see in the method section) showed significantly larger activation in the AC block compared to the SOM block (Table 4 and Figure 3): the left IFG (left IFG: X = −42, Y = 4, Z = 30; mean beta for AC = 0.367; for SOM = 0.166; M *diff* = 0.2, SD = 0.26, t(18) = 3.30, *p* < 0.01) and the left IPL (X = −44, Y = −40, Z = 42; mean beta for AC = 0.28; for SOM = 0.08; M *diff* = 0.2, SD = 0.33, t(18) = 2.68, *p* < 0.01).

Next, we looked for correlations between the brain activations at each ROI and the behavioral performances on the tasks. Interestingly, we found a significant correlation between the accuracy rate and the brain activation in one ROI: the left IFG (Table 4 and Figure 3). A positive correlation was found in this ROI for both AC block and SOM blocks (for AC block: r = 0.532, *p* < 0.05; for SOM block: r = 0.487, *p* < 0.05), with no significant difference between those correlations (Fisher’s z = 0.172, *p* = 0.432). None of the other ROIs showed a significant correlation with the accuracy rate either for the AC block or for the SOM block (see Table 4).

## 4. Discussion

The present study aimed at examining the neural activation underlying performance in the computation estimation task requiring participants to estimate the magnitude of the results of a multi-digit multiplication problem relative to a given standard. We looked at the neuronal correlates of the two most common strategies used in computational estimation: (1) the AC strategy involves rounding and multiplication procedures [18,27,31]; (2) the SOM strategy is based on intuitive approximate magnitude representations for the results, without any calculation. To test the neural network that associates with each of these strategies, during an fMRI scan, we asked participants to solve the computation estimation task in the no-choice paradigm: in one block using AC strategy only and in the other block using the SOM strategy only.

Behaviorally, in line with previous studies, the SOM block produced faster but less accurate responses than the AC block [23,24,30,32,33].

Brain-wise, the two blocks (SOM and AC) invoked multiple brain regions related to calculation, numerical cognition, decision making, and working memory. The AC block engaged regions in the frontal cortex, specifically the middle and inferior frontal gyrus and the insula. Moreover, as expected, we found extensive activation in the parietal cortex, including the IPS and in the fusiform gyrus.

The SOM block extensively invoked the frontal cortex, including the middle and superior frontal gyrus. Moreover, as expected, we found activation in the parietal cortex, including the right SMG, and the left AG in the IPL. We also discovered activations in the (1) bilateral posterior cingulate cortex; (2) occipital lobule, specifically the inferior and the lateral occipital gyrus; and (3) bilateral temporal gyrus.

Direct comparison of the two blocks revealed that the AC block invoked greater activation than the SOM block in the frontal cortex, including the bilateral inferior frontal gyrus, right superior and middle frontal, left orbital gyrus, and right insula. These neural differences are probably due to greater involvement of working memory and task difficulty demands in the AC than in the SOM blocks. Specifically, the AC requires calculations of rounded operations, while the SOM does not require any calculations, reflecting differentiate difficulty levels.

However, we also discovered greater frontal activity for the reverse contrast (SOM more than AC), including the bilateral superior and inferior frontal gyrus. These findings indicate that each of the strategies require differentiated domain general demand, such as specific working-memory systems. Furthermore, the SOM activated the cingulate gyrus, probably due to decision making, which is needed more in SOM than approximate calculation. The cingulate gyri have been associated with error monitoring [34], integration of information [35], and resolving interference, such as in a Stroop task [36].

For the contrast of AC more than SOM, we discovered multiple, domain-specific brain regions in the parietal lobule, including the left SMG (BA 40), the bilateral SPL (BA 7), and the right IPL (BA 7). These findings suggest that the analogic code of magnitude is more involved in the AC block than in the SOM block [7,37].

Activations in the contrast of SOM more than AC were also found in the parietal lobule, including the right SMG (BA 39). The right SMG was found to be related to visuospatial working-memory demand [38]. We suspect that the SOM strategy involves activation of a mental number line. In the SOM strategy, an approximate result and the reference number of the computational estimation problem should be located on a mental number line spatially. The spatial process requires visuospatial working memory; hence, it engage the right SMG.

ROI analysis was performed on two predefined regions in the left hemisphere that showed greater activation in the AC than the SOM block: IFG and IPL. Additionally, we found that the activation level of the left IFG was positively correlated to individual accuracy in the AC and SOM blocks, thus indicating that the left IFG has an essential role in both strategies.

### 4.1. Distinct Codes Are Involved in SOM and Approximate Calculation

The “triple-code: model predicts that numbers are processed in three numerical surface formats: (1) a visual Arabic code represented by strings of digits; (2) an analogic quantity and magnitude code activated indirectly by one of the other codes, possessing size information on a mental number line; and (3) a verbal code represented by words [39]. These codes are based on distinct brain areas: (1) bilateral activity in inferior ventral occipito-temporal areas underlying the visual Arabic code; (2) activity in inferior parietal areas underlying quantity and magnitude judgments. The IPS was suggested to host a core quantity system analogous to an internal “number line” [40]; and (3) the left peri-sylvian areas and the left AG underlying a verbal code [40,41]. In line with this anatomical distinction, the present results found that activation during the SOM strategy block is located in brain regions related to the visual Arabic code. Specifically, the SOM block had greater activation than the AC block in the (1) occipital lobule, including the bilateral superior occipital gyrus, right inferior occipital gyrus, bilateral lingual gyrus, and the left cuneus, and (2) temporal lobule, including bilateral middle and superior-temporal gyrus and the left inferior temporal gyrus. The neural network associated with the visual Arabic code processes symbolic Arabic numerals without linking the numerals to their full semantic meaning.

The semantic code is needed in both SOM and in AC strategy. However, due to the greater involvement of approximation processes in SOM than in AC, we hypothesized greater engagement of the IPS in SOM than in the AC. On the contrary, we found greater bilateral invocation in the SPL near the IPS for the AC than the SOM. Similarly, in ROI analysis, we targeted the location of the IPS according to meta-analysis [7] and did not discover any difference in the activity level of the IPS between the two strategies. Moreover, looking at the activation of each strategy compared to baseline suggests that brain activation for AC but not the SOM was found in the IPS. Hence, based on the brain imaging result and contrary to the predictions of the present study, we suggest that involvement of domain specific parietal regions is higher in AC than in the SOM blocks.

Exact calculation, which is involved more in AC than in SOM strategy, is associated with the verbal code. Hence, in line with the anatomical location of the verbal code, we hypothesized that left inferior parietal regions will be more engaged in the AC than the SOM strategy. Even though we could not find an indication for greater engagement in the left AG in AC, we discovered that the left SMG, located in the left inferior parietal lobe, was shown to be engaged more in AC as compared to the SOM block. The left SMG was found to be activated during mental calculations [42] and calculating the price after discount [43], and its activation is related to verbal working-memory demand [41]. Specifically, the left SMG was found to be related to verbalization during mental arithmetic but not to visual representation [39]. The AC requires multiple verbal steps of solution: first, rounding both operations up or down, multiplying them, and then comparing the result to the reference number. The SOM strategy requires fewer verbal steps. Accordingly, the SOM strategy requires less verbal working-memory demand than the AC strategy.

### 4.2. Left Lateralization of Brain Activity in AC but Not SOM

Even though we tested the ROI located symmetrically in the left and right hemispheres, only ROIs located on the left but not right were found to have greater invocation in the AC than the SOM (L IFG and L IPL). This finding, along with the whole-brain-level findings, indicates that AC strategy involves a leftward lateralization of brain activity. This pattern of asymmetry was unique to the AC block and was not observed during the SOM block. The triple-code model [35,40] suggests that bilateral IPS are responsible for quantity manipulations, while exact verbal calculations are more strongly represented in the left hemisphere (mostly in the left AG). However, multiple later studies have discovered leftward lateralization of brain activity in the IPS (not the AG) during symbolic number processing [44,45,46]. One main difference between the activation of the left and right IPS is related to the involvement of exact calculation compared to approximation. Exact calculation is associated with engagement of the left IPS, while approximation is associated with the right IPS [9,38]. Looking at the present study, although AC and SOM both involve approximation, AC requires direct calculation (multiplication after rounding the operations up or down), while SOM does not require any calculation. Accordingly, left lateralization of brain activity can be related to involvement of calculations in the AC strategy.

## 5. Conclusions

In the no-choice paradigm, we examined the neuronal correlates of two strategies used in computational estimation: (1) the AC strategy involves rounding and multiplication procedures; (2) the SOM strategy is based on intuitive approximate magnitude representations for the results without any calculation. Behaviorally, the SOM strategy produced faster but less accurate responses than the AC strategy.

The two blocks (SOM and AC) were associated with changes in neural activity in multiple brain regions related to calculation, numerical cognition, decision making, and working memory.

The AC more than the SOM block elicited activations in multiple, domain-specific brain regions in the parietal lobule, including the left SMG (BA 40), the bilateral superior parietal lobule (BA 7), and the right IPL (BA 7). These findings suggest that the analogic code of magnitude is more involved in the AC block than in the SOM block.

ROI analysis discovered two predefined regions in the left hemisphere that showed greater activation in the AC than the SOM: IFG and IPL. These findings from the ROI analysis support results from the whole-brain-level analysis: greater involvement of the fronto-parietal network in AC than in the SOM block. In the ROI analysis, we also discovered that the activation level of the left IFG was positively correlated with individual accuracy in the AC and SOM blocks. This finding indicates that the left IFG has an essential role in both strategies.

## Figures and Tables

**Figure 1 brainsci-12-00357-f001:**
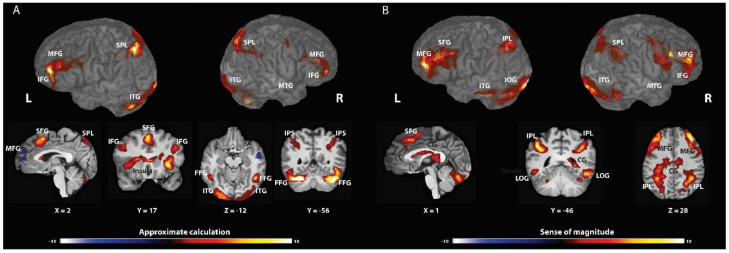
Brain activations associated with approximate calculation strategy (AC-A) or sense of magnitude (SOM-B). (**A**) AC elicited activation in the frontal cortex; in the posterior parietal cortex, including the intraparietal sulcus, and the fusiform gyrus. Significant deactivations were found in the right middle temporal gyrus and in the left medial frontal gyrus. (**B**) SOM invokes regions in the frontal cortex; in the posterior parietal cortex in the right and left posterior cingulate cortex; in the occipital lobule; and in the left inferior temporal gyrus.

**Figure 2 brainsci-12-00357-f002:**
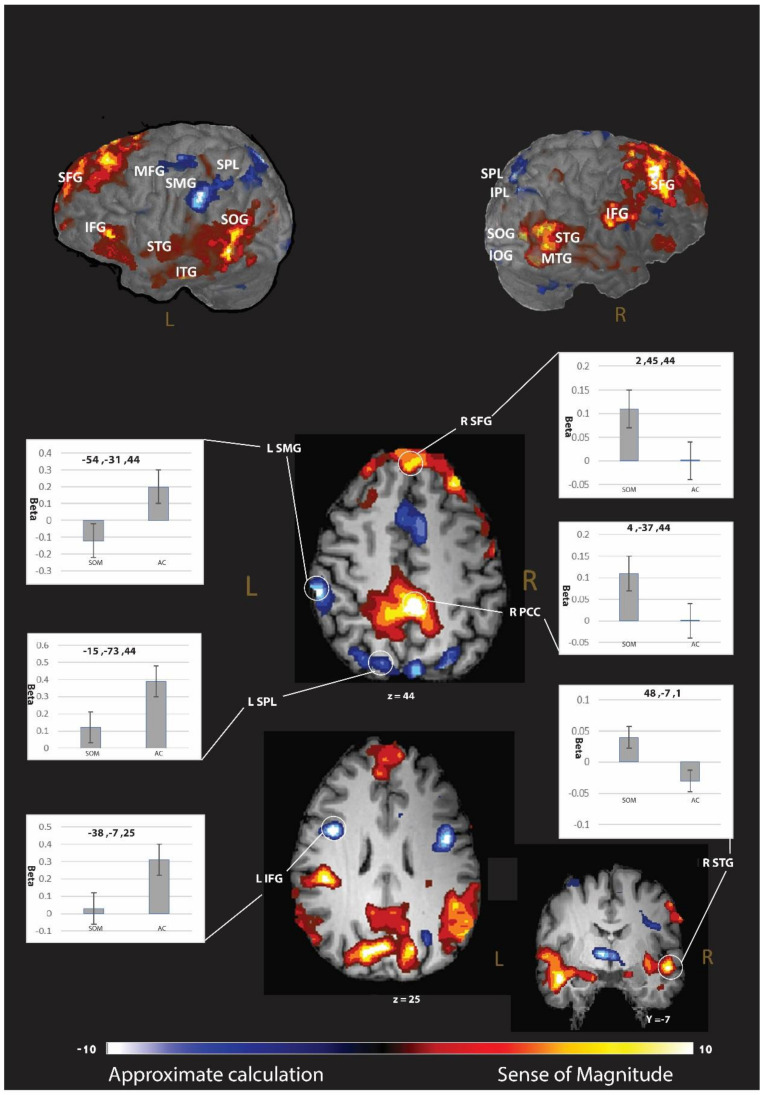
Brain regions associated with changes in neural activity in approximate calculation strategy related to sense of magnitude strategy. To unravel the brain regions that showed significant brain activation differences between the strategies, we calculated a whole-brain *t*-test between the brain activation of the two strategies.

**Figure 3 brainsci-12-00357-f003:**
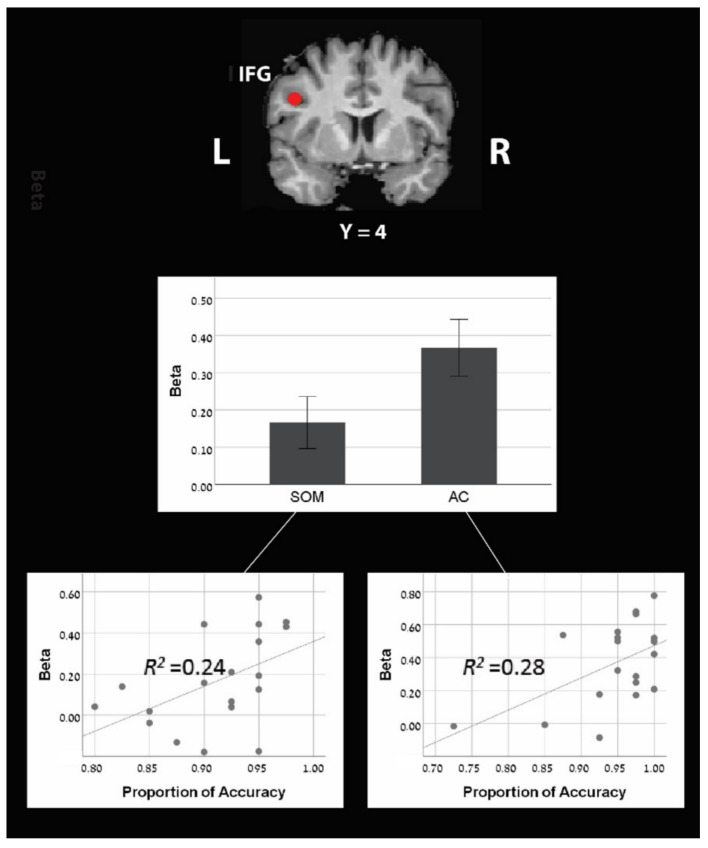
Region of interest analysis of the left IFG (X = −42, Y = 4, Z = 30) shows greater activation during the approximate calculation strategy compared to the sense of magnitude strategy. Additionally, a positive correlation was found between accuracy in computational estimation and activity level both for AC and SOM blocks.

**Table 1 brainsci-12-00357-t001:** Brain areas that showed significant engagement in approximate calculation block.

Region	Hem.	BA	Number of Voxels	Peak Talairach Coordinates			Peak t(18) Score
				X	Y	Z	
Activations							
Middle frontal gyrus	R	9	26,647	36	35	31	10.245
Insula	R			29	20	8	9.544
Middle frontal gyrus	L	9	14,114	−40	36	26	6.980
Inferior frontal gyrus	L	46		−39	43	8	6.427
Inferior frontal gyrus	L	6		−35	1	25	6.439
Intraparietal sulcus	L	7	20,329	−32	−66	48	8.822
Intraparietal sulcus	L	19		−29	−66	36	8.535
Cuneus	L	7		−9	−72	37	8.061
Intraparietal sulcus	L	40		−49	−43	44	7.994
Cuneus	R	7		8	−77	45	6.358
Intraparietal sulcus	R	7		28	−61	30	5.255
Superior frontal gyrus	L	8	5669	−2	15	49	7.396
Insula	L		5651	−28	11	9	4.963
Extra-nuclear/Corpus clausum	L	30	1367	−20	−39	8	6.480
Fusiform gyrus	R	37	46,314	48	−52	−14	8.160
Fusiform gyrus	L	18		−29	−85	−14	7.571
Fusiform gyrus	L	37		−48	−49	−17	6.706
** *Deactivations* **							
Middle temporal gyrus	R	22	1107	51	−7	−8	6.259
Medial frontal gyrus	L	10	2189	−2	51	13	5.436

**Table 2 brainsci-12-00357-t002:** Brain areas that showed significant activation in sense of magnitude block.

Region	Hem.	BA	Number of Voxels	Peak Talairach Coordinates			Peak t(18) Score
				X	Y	Z	
*Activation*							
Middle frontal gyrus	R	10	23,252	36	41	25	9.477
Middle frontal gyrus	R	6		33	−1	52	6.369
Middle frontal gyrus	L	10	22,750	−36	41	22	10.062
Superior frontal gyrus	L	6	3684	−1	14	46	6.098
Supramarginal gyrus	R	40	49,677	45	−46	37	8.744
Inferior parietal lobule	L	40		−42	−46	37	6.980
Inferior parietal lobule	L	7		−36	−64	43	6.450
Posterior cingulate cortex	R	23		6	−10	25	6.380
Sub-gyral *	L			−30	−28	28	5.850
Posterior cingulate cortex *	L			−12	−13	25	5.045
Insula	R	13	4119	36	11	4	6.207
Inferior occipital gyrus	L	18	69,627	−30	−87	−8	10.112
Inferior temporal gyrus	L	21		−57	−28	−14	8.490
Lateral occipital gyrus	L	37		−39	−61	−14	7.360
Inferior occipital gyrus	R	18		21	−91	−13	7.183
Lateral occipital gyrus	R	37		48	−52	−11	6.394
*Deactivation*							
None							

* Regions that were no longer significant at the higher threshold (α < 0.001), see Appendix A.

**Table 3 brainsci-12-00357-t003:** Brain areas that showed significant differences between approximate calculations compared to sense of magnitude block.

Region	Hem.	BA	Number of Voxels	Peak Talairach Coordinates			Peak t(18) Score
				X	Y	Z	
AC strategy > SOM strategy							
Inferior frontal gyrus	R	6	13,160	40	1	23	8.818
Superior frontal gyrus	R	6		7	3	49	7.542
Cingulate gyrus	R	32		8	16	32	6.755
Cingulate gyrus *	L	24		−10	10	32	4.430
Middle frontal gyrus	R	6		28	−6	52	4.050
Insula *	R		3658	29	21	8	5.959
Supramarginal gyrus	L	40	12,888	−54	−31	44	8.569
Cuneus	R	7		5	−77	45	8.358
Superior parietal lobule	R	7		24	−73	41	6.821
Inferior parietal lobule	R	7		26	−63	27	6.810
Superior parietal lobule	L	7		−15	−73	44	6.160
Middle occipital gyrus	R	17	3689	21	−92	−9	7.413
Thalamus	L		3623	−7	−5	4	8.260
Thalamus	R			6	−5	2	7.120
Inferior frontal gyrus	L	9	2017	−38	7	26	10.228
Lateral occipitotemporal gyrus	L	37	643	−43	−54	−14	6.857
Inferior occipital gyrus	L	18	4054	−23	−87	−9	5.952
Orbital Gyrus *	L	10	970	−30	47	−4	5.301
Precentral gyrus	L	6	1561	−30	−7	59	4.813
SOM strategy > AC strategy							
Superior frontal gyrus	R	6	33,786	2	45	37	11.022
Superior frontal gyrus	R	8		23	27	52	10.350
Superior frontal gyrus	L	8		−20	30	51	7.452
Postcentral gyrus	R	4		54	−11	38	6.225
Cingulate gyrus	R	31	183,586	4	−37	40	10.296
Superior occipital gyrus	R	18		14	−76	18	10.167
Middle temporal gyrus	L	21		−44	−7	−15	9.633
Superior temporal gyrus	L	42		−53	−36	12	9.229
Superior occipital gyrus	L	18		−18	−83	16	9.214
Cuneus	L	30		−19	−68	13	9.150
Lingual gyrus	R			22	−42	−10	8.937
Lingual gyrus	L			−7	−67	−5	8.445
Precuneus	L	31		−13	−37	36	8.392
Inferior temporal gyrus	L	21		−48	−5	−22	7.419
Inferior occipital gyrus	R	37		51	−64	0	6.881
Supramarginal gyrus	R	39		55	−55	22	6.760
Superior temporal gyrus	R	22		48	−32	1	6.728
Middle temporal gyrus	R	21		50	−46	5	5.860
Inferior frontal gyrus	R	46	1198	48	31	8	6.565
Inferior frontal gyrus	L	45	4984	−48	34	5	9.761
Inferior frontal gyrus	L	46		−50	31	18	9.276

* Regions that were no longer significant at the higher threshold (α < 0.001), see Appendix A.

**Table 4 brainsci-12-00357-t004:** Region of interest analysis.

Source	ROI	Talairach Coordinates			Mean Differences (AC Strategy –SOM Strategy)	*t* (18) Score	*p*-Value *(Right-Tail)*	Correlation with Accuracy (and *p*-Value)	
		X	Y	Z				AC Block	SOM Block
Arsalidou and Taylor, 2011	Right IFG	46	10	28	0.07	1.02	0.16	0.148 (0.32)	0.05 (0.56)
	Left IFG	−42	4	30	0.20 *	3.30	0.002	0.53 * (0.02)	0.49 * (0.03)
	Right IPL	46	−34	46	−0.02	−0.34	0.36	0.18 (0.45)	−0.01 (0.90)
	Left IPL	−44	−40	42	0.20 *	2.68	0.007	0.30 (0.20)	0.20 (0.40)
Cohen Kadosh et al., 2008	Right IPS	36	−49	42	0.05	0.54	0.29	0.18 (0.47)	−0.02 (0.92)
	Left IPS	−32	−47	47	−0.01	−0.09	0.46	0.14 (0.55)	0.19 (0.41)

IFG, inferior frontal gyrus; IPL, inferior parietal lobe; IPS, intraparietal sulcus. * Regions that were no longer significant at the higher threshold (α < 0.001), see Appendix A.

## Data Availability

Data will be given upon request from the first author.

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
