# Peer review of "Neural Correlates of Numerical Estimation: The Role of Strategy Use"

_brainsci, 2022, doi:10.3390/brainsci12030357_

Round 1

Reviewer 1 Report

In current study the authors addressed an interesting topic - the neural correlates underlying computation estimation. Authors aimed to explore the neuronal correlates of two strategies, approximated calculation and sense of magnitude. Although they present some results on this topic as mentioned above, I still have questions about the study. I would elaborate my questions as follows.

Major issues:

  1. In the introduction part, the authors introduced the previous studies in calculation, exact and approximate calculation, and computation estimation by listing the fMRI results. It is important to reorganized the previous work focusing on scientific questions instead of brain activation.
  2. In”Present study”part, authors would better to put forward their hypothesis on the neural mechanism of computation estimation, or author would show the reasons why they believed they would get such results.
  3. In materials and methods part, authors said “Twelve participants were excluded from the analysis due to movement greater than 1mm during the scan” in line 161, while in line 271 they described that“Five subjects were excluded in 217 this stage due to extreme head movements inside the magnet (> 2mm) “ with a confutation.
  4. According the experiment design, each block had 40 trails with a jitter alternation(10, 12, 16 seconds). It is a classical event-related design, and the beta values of AC/SOM strategy condition could be conducted at the first individual level. Then, the group activation map for each condition, and AC vs. SOM or SOM vs. AC activation maps were conducted in the second level analysis. But, in the study, authors used one block (AC/SOM) vs. rest, which confused me, as well as the processing “the time window for the fMRI analysis was 6s”in line 199.
  5. In results part, in Fig 3, the results were not sufficiently robust because the values of R2 was small, additionally, the effect of correlation might be caused by the lowest accuracy.

Author Response

Reviewer 1

  1. In the introduction part, the authors introduced the previous studies in calculation, exact and approximate calculation, and computation estimation by listing the fMRI results. It is important to reorganized the previous work focusing on scientific questions instead of brain activation.

We have reorganized the introduction, it now includes more details regarding the scientific issues.

  1. In” Present study” part, authors would better to put forward their hypothesis on the neural mechanism of computation estimation, or author would show the reasons why they believed they would get such results.

We have justified our predictions in the present study and reorganized the present study section in the introduction.

  1. In materials and methods part, authors said “Twelve participants were excluded from the analysis due to movement greater than 1mm during the scan” in line 161, while in line 271 they described that “Five subjects were excluded in 217 this stage due to extreme head movements inside the magnet (> 2mm) “ with a confutation.

Thank you for drawing our attention to this mistake. The dataset we used in this study is part of a larger dataset containing both participants that performed the task during the fMRI (24 participants, for whom we have behavioral and fMRI data) and participants who participated only in the behavioral part (7 participants). Here we focused only on the brain-activity aspect of that dataset, that is, the fMRI data from the 24 participants. Five participants out of 24 were excluded due to extreme head movement (>2mm), as mentioned in the paper (line 217). We corrected the text accordingly, on line 161..

  1. According the experiment design, each block had 40 trails with a jitter alternation(10, 12, 16 seconds). It is a classical event-related design, and the beta values of AC/SOM strategy condition could be conducted at the first individual level. Then, the group activation map for each condition, and AC vs. SOM or SOM vs. AC activation maps were conducted in the second level analysis. But, in the study, authors used one block (AC/SOM) vs. rest, which confused me, as well as the processing “the time window for the fMRI analysis was 6s” in line 199.

We performed two second level analyses: 1) comparing each strategy individually to the rest (see 3.2.1 and 3.2.2). 2) Comparing AC vs. SOM or SOM vs. AC activation maps (see 3.2.3). The purpose of the first was to discover brain areas that were involved in each strategy. The purpose of the second was to discover differences between the strategies at the neural level. Here, we no longer need the ‘rest’ neural activities, as we can directly compare both strategies’ neural activations.

We also clarified the protocol and task design, see line 208, p. 5: "At the beginning of each block, there was a blank slide for 30 seconds, and at the end of each block for 60 seconds. A trial contained a 2-digit × 2-digit multiplication problem with a target number below the operation, the participants were asked to indicate, by a keypress, whether the exact solution was larger or smaller than the target number. The problem and the target number appeared for 12 seconds. After each trial, there was a jitter alternation between 10, 12, or 16 seconds randomly. Each trial in our fMRI protocol was set for 12 seconds from the appearance of the multiplication problem until the disappearance of that problem." 

  1. In results part, in Fig 3, the results were not sufficiently robust because the values of Rwas small, additionally, the effect of correlation might be caused by the lowest accuracy.

According to Cohen’s d, the correlation values were medium, and each of them was significant with a small sample size for correlations (n= 19). Specifically, the R2 scores were 0.24 and 0.28, which are equivalent to the Pearson’s r scores of 0.49 and 0.53 respectively. Regarding the second issue, the effect of correlation might be caused by the lowest accuracy. We recalculated the correlation between accuracy and activation level, omitting one participant with the lowest accuracy (“outlier” observation - participant who achieved 0.75 accuracy). In this case, the r score dropped from 0.53  to 0.43, which reflects a fairly high association. The p-value of the new r-score (p=0.06) approaches significance, but can be attributed to the smaller N. We are not certain that the participant that we excluded is a real outlier. Hence, we would like not to include that analysis in the text. However, if the reviewer recommends it, we can include the analysis in the main text or the supplementary material.

Reviewer 2 Report

In their work, Ashkenazi, Tikochinski and Ganor-Stern looked at the neural correlates of approximate calculation, and (more intuitive) sense of magnitude, two disparate strategies for evaluating “numerosity”. Approximate calculation involved more wide-spread neural network, including the left supramarginal gyrus, and right precuneus. The neural activity in the inferior frontal gyrus was positively correlated with accuracy in both tasks.

This neuroimaging contribution to literature on computation estimation is quite valuable, and in general, the manuscript is rather well written, and quite interesting. Yet, in its current form there are a few issues that should be addressed by the authors.

Throughout the manuscript text, while referring to the observed neural activity (or even the outcomes of earlier studies), the authors almost always use the term "activation", and say that tasks "activated". Depending on the brain region, and tasks, oftentimes in the control condition an areas is suppressed, resulting in significant differences between the experimental and control tasks, but not necessarily in "activation" of a given area. While such subtle distinction may not be critical for all reported findings, the alternative phrases include "engage", "invoke", "are associated with changes in/of neural activity in", with the latter phrase also weakening the issue of causation, which is inherently present there, as well.

In a few places, the concept of “neurological” (evidence, basis, etc.) is used. Yet, it is best suited when referring to patients (with brain damage, as on p. 3, line 141), but not necessarily to healthy participants. (Why not neural? or neuronal?)

Thirty-one participants were tested, but as many as twelve were excluded due to excessive head movements. While 19 participants is still a decent number to test neuroimaging hypotheses, there is one striking thing in the description of methods. P. 5, line 217-18: “Five subjects were excluded in this stage due to extreme head movements inside the magnet (> 2mm)”. There was no further info on why the remaining seven were excluded. Yet, inferring logically from what is written, they had head movements smaller than 2 mm. If this is the case, and there was only a gradual drift in (rather than abrupt and numerous) head displacements smaller than 2 mm, removing these participants from analyses may not be necessary. But too little (actually nothing) is known about these cases.

Along the same lines, the “height threshold” for statistical analyses was set at p < 0.005 (t18 > 3.???, two-sided; these values are not provided). Yet, whatever the value is, it is definitely lower than the recommended (normalized) Z value of 3.1 (Eklund, A., Nichols, T.E., & Knutsson, H. (2016). Proc Natl Acad Sci U S A, 113(28), 7900-7905. doi: 10.1073/pnas.1602413113). On the one hand, your threshold should already give reliable results. Yet, a threshold of t18 > 3.6 should still give pretty much the same results (judging by the activity patterns presented).

Minor:

In Figure 1, contrary to the provided descriptions in figure caption and text, I cannot find any neural activity in the right middle temporal gyrus.       

P. 12, line 342: the right SMG, and the left IPL? (SMG is also part of IPL)

P. 13, line 370: “ROI analysis revealed two predefined regions in the left hemisphere that showed” sounds strange. It should probably read “performed in (instead of ‘revealed’) … showed that …”

P. 14, line 407: “The left SMG was found to be activated during mental calculations [47], and” – in addition to replacing “was found to be activated during” with “was shown to be engaged more in”, I would recommend adding a reference to a recent paper by Klichowski and collaborators: Klichowski et al. (2020). Mental shopping calculations: A transcranial magnetic stimulation study. Frontiers in Psychology, 11(1930), 1-7. https://doi.org/0.3389/fpsyg.2020.01930. It seems to be quite relevant, at any rate.

Author Response

  1. Throughout the manuscript text, while referring to the observed neural activity (or even the outcomes of earlier studies), the authors almost always use the term "activation", and say that tasks "activated". Depending on the brain region, and tasks, oftentimes in the control condition an areas is suppressed, resulting in significant differences between the experimental and control tasks, but not necessarily in "activation" of a given area. While such subtle distinction may not be critical for all reported findings, the alternative phrases include "engage", "invoke", "are associated with changes in/of neural activity in", with the latter phrase also weakening the issue of causation, which is inherently present there, as well. In a few places, the concept of “neurological” (evidence, basis, etc.) is used. Yet, it is best suited when referring to patients (with brain damage, as on p. 3, line 141), but not necessarily to healthy participants. (Why not neural? or neuronal?)

Thank you for this comment, we changed the terminology in all of the manuscript.

  1. Thirty-one participants were tested, but as many as twelve were excluded due to excessive head movements. While 19 participants is still a decent number to test neuroimaging hypotheses, there is one striking thing in the description of methods. P. 5, line 217-18: “Five subjects were excluded in this stage due to extreme head movements inside the magnet (> 2mm)”. There was no further info on why the remaining seven were excluded. Yet, inferring logically from what is written, they had head movements smaller than 2 mm. If this is the case, and there was only a gradual drift in (rather than abrupt and numerous) head displacements smaller than 2 mm, removing these participants from analyses may not be necessary. But too little (actually nothing) is known about these cases.

Thank you for bringing it to our attention, there were 24 participants (not 31 participants) and 5 were excluded. We have corrected the text accordingly. See also our response to issue no.3 of reviewer 1.

  1. Along the same lines, the “height threshold” for statistical analyses was set at p < 0.005 (t18 > 3.???, two-sided; these values are not provided). Yet, whatever the value is, it is definitely lower than the recommended (normalized) Z value of 3.1 (Eklund, A., Nichols, T.E., & Knutsson, H. (2016). Proc Natl Acad Sci U S A, 113(28), 7900-7905. doi: 10.1073/pnas.1602413113). On the one hand, your threshold should already give reliable results. Yet, a threshold of t18 > 3.6 should still give pretty much the same results (judging by the activity patterns presented).

Following the reviewer’s comment, we re-ran the analyses, applying a higher threshold (t(18)>3.6, p<0.001). As the reviewer predicted, the overall results indeed remained the same. All the brain areas of three contrast maps (AC-rest, SOM-rest, AC-SOM) were also found in the new analysis, except for five minor areas (two areas in the SOM-rest contrast and three areas in the AC-SOM contrast). In addition, as expected, the cluster-size (i.e., the number of voxels of the area) of a couple of brain areas were slightly reduced. We have provided detailed information in three new supplementary tables (corresponding to Tables 1-3 of the paper) so the reader can compare the results. To inform the reader about the five brain-areas who omitted after the new threshold, we added an asterisk next to these areas in the paper’s tables.

  1. In Figure 1, contrary to the provided descriptions in figure caption and text, I cannot find any neural activity in the right middle temporal gyrus.

Figure 1 includes two views of the MTG, please see the deactivation (in blue) in z= -12 and the whole brain right side, the activations are small but are present.

  1. 12, line 342: the right SMG, and the left IPL? (SMG is also part of IPL)

Thank you, we have changed this in the text.

  1. 13, line 370: “ROI analysis revealed two predefined regions in the left hemisphere that showed” sounds strange. It should probably read “performed in (instead of ‘revealed’) … showed that …”

Done.

  1. 14, line 407: “The left SMG was found to be activated during mental calculations [47], and” – in addition to replacing “was found to be activated during” with “was shown to be engaged more in”, I would recommend adding a reference to a recent paper by Klichowski and collaborators: Klichowski et al. (2020). Mental shopping calculations: A transcranial magnetic stimulation study. Frontiers in Psychology, 11(1930), 1-7. https://doi.org/0.3389/fpsyg.2020.01930. It seems to be quite relevant, at any rate.

Done, we have added this reference to the manuscript.

Round 2

Reviewer 1 Report

The MS was improved.  But the sample size is small, and I'd suggest authors to add subjects if you can.